# Kiwifruit Detection Method in Orchard via an Improved Light-Weight YOLOv4

Jianbo Gao [1], Sihui Dai [2], Jingjing Huang [3], Xu Xiao [4], Li Liu [5], Longhan Wang [1], Xia Sun [1], Yemin Guo [1] and Ming Li [1,3,*]

1 School of Agricultural Engineering and Food Science, Shandong University of Technology, Zibo 255000, China
2 School of Horticulture and Landscape, Hunan Agricultural University, Changsha 410128, China
3 Hunan Agricultural Equipment Research Institute, Changsha 410125, China
4 School of Electrical and Information Engineering, Hunan University, Changsha 410128, China
5 School of Foreign Languages (Sdut), Shandong University of Technology, Zibo 255000, China
* Correspondence: liming@hunau.net

**Abstract:** In order to detect kiwifruit quickly and accurately in orchard environments for the picking robot, this paper proposed a detection method based on a lightweight YOLOv4-GhostNet network. The implementations of the method are as follows: The original CSP-Darknet53 backbone network model was replaced by GhostNet, a feature layer facilitating small object detection was introduced in the feature fusion layer, and part of the ordinary convolution was replaced by a combination of $1 \times 1$ convolution and depth-separable convolution to reduce the computational pressure caused by the fused feature layer. The parameters of the new network are reduced, and the generalization ability of the model is improved by loading pre-training weights and freezing some layers. The trained model was tested, and the results showed that the detection performances were better than that of the original YOLOv4 network. The F1 value, map, and precision were improved on the test set, which were 92%, 93.07%, and 90.62%, respectively. The size of weight parameters was reduced to 1/6 of the original YOLOv4 network, and the detection speed reached 53 FPS. Therefore, the method proposed in this study shows the features of fast recognition, lightweight parameters, and high recognition accuracy, which can provide technical support for vision systems of kiwifruit picking robots.

**Keywords:** kiwifruit detection; YOLOv4; GhostNet; light-weight; picking robot





## 1. Introduction

Kiwifruit is one of the most productive fruits in China and has great economic benefits. Because of the increasing cost of manual picking in recent years, many agricultural industries have to use picking robots [1]. A vision system is a vital part of vision-based picking robots, affecting performances such as efficiency, stability, and adaptation in complex environments [2,3]. However, there are factors that make the vision system unstable: the varying light intensity due to changing weather conditions; the diversity of fruit clusters with different branch, leaf shading, and overlapping; the limited computational resources and complex algorithms which do not run efficiently for the picking robot. These factors make it difficult for picking robots to quickly and accurately detect kiwifruit.

In recent years, researchers around the world have conducted lots of studies on the object recognition of fruit and vegetables in natural environments, including traditional image processing techniques and deep learning methods which are currently popular. Traditional recognition methods mainly include edge contour extraction methods, region growth segmentation methods, threshold segmentation methods, etc. The fruit objects in images are usually recognized by using single feature or the combination of multiple features extracted from the fruit images, such as shapes, textures, and color differences. Hussin et al. [4] used a circular Hough transform method for citrus object detection, but the detection accuracy was low for dense and overlapping fruits. Payne et al. [5] employed RGB

and YCbCr color space segmentation as well as texture segmentation based on adjacent pixel variability to segment mangoes from background pixels. Sun et al. [6] proposed a string harvest tomato segmentation method based on the Canny edge detection algorithm, which solved the problem of fruit adhesion, but also wasted a large number of non-fruit adhesion points. Scarfe et al. [7] used the Sobel edge algorithm to remove the target fruit background and identify kiwi using the template matching method, but did not use the fruit shape information. Peng et al. [8] presented methods such as shape invariant moments to synthesize the color and shape features of fruits and used an SVM classifier to classify fruits, but the applicability of the algorithm is relatively poor for different environments. The methods mentioned above can identify a single type of fruit, but they are poorly adapted to situations such as similar color backgrounds, fruit shading, and light changes, resulting in poor generality. So, traditional machine vision technologies are limited by their classification algorithms and cannot meet the requirements for picking robots in complex environments [9].

Deep learning object detection algorithms can complete the fruit recognition tasks quickly and reach great performances, which are mainly divided into two categories: the first one is the regression-based one-stage object detection algorithm, including YOLO [10], Single Shot MultiBox Detector (SSD) [11], etc., and the other is the two-stage algorithm based on region suggestion, the representative algorithms include Faster RCNN [12], RCNN [13] and Mask R-CNN [14], etc. Sa et al. [15] and Song et al. [16] used Faster R-CNN networks to identify bell pepper and kiwi, respectively and improved the network recognition accuracy. However, the two-stage object detection algorithm model is slower to train and has a longer detection time than that of one-stage algorithms. Fu Longsheng et al. [17] proposed a LeNet convolutional neural network-based multi-cluster kiwifruit recognition method, which used elevated angle of capture for image acquisition of trellis cultivated kiwifruit, with high accuracy for independent and adjacent fruit recognition. However, the accuracy is relatively low for obscured and overlapping fruits and the recognition speed is slow for individual fruits. Tian et al. [18] presented an improved YOLOv3 network using DenseNet as its feature extraction layer for detecting apples at different growth stages, but lacked fruit recognition in large-view scenes. Lu et al. [19] proposed a lightweight neural network based on an improved YOLOv3-LITE using MobileNetv2 as the backbone of the model, which has an average accuracy of 91.13% and a recognition speed of 16.9 ms for a single image on a computer workstation. Fu et al. [20] proposed an application for kiwi picking by improving YOLOv3-tiny lightweight neural network for robot object detection, which has a model weight of 27 MB and an average detection speed of 34 ms per image on a robotic workstation, but YOLOv4 has a better balance of detection accuracy and detection speed than YOLOv3. Based on the kiwi occlusion, Suo et al. [21] used YOLOv3 and YOLOv4 to classify the target fruits into multiple classes for detection, and the results showed that the highest mAP of 91.9% was achieved by YOLOv4, which cost 25.5 ms on average to process an image, but no improvement was made to the original YOLOv4 network. The original YOLOv4 network structure is too large, with high computational complexity and huge model size, which is not suitable for deployment in picking robots for real-time detection [22–24].

To achieve fast and precise identification of kiwifruit picking robots in the case of scaffolding cultivation of kiwifruit with insufficient light, overlapping, and clustering, an improved lightweight GhostNet-YOLOv4 neural network is proposed. A feature fusion layer is introduced which is favorable for small object detection, and a combination of $1 \times 1$ convolution and depth-separable convolution is introduced to achieve the function of ordinary convolution by borrowing the Ghost Module structure, compressing the number of network parameters and improving the detection speed of the network. To verify the effectiveness of the improved object detection algorithm in detection, the experiments of the improved algorithm in different scenarios and the results are compared with four classic object detection algorithms: SSD, YOLOv3, YOLOv4, and MobileNetV3-YOLOv4.

## 2. Materials and Methods

### 2.1. Image Data Acquisition

The kiwifruit images were acquired at the orchard base of Hunan Academy of Agricultural Sciences at different times of the day. The camera used for acquisition was a RGB-D camera, RealSense D435i, which is manufactured by Intel (Santa Clara, CA, USA). The camera was positioned 20–90 cm away from the kiwifruit fruit for the acquisition, as shown in Figure 1. To simulate the vision system of a picking robot, different elevation angles were used for the shots, and a total of 2325 raw kiwifruit images were collected. To avoid overfitting phenomena due to insufficient diversity of sample data acquisition, the presence of branches and leaves shading, fruit adjacency and denseness of scaffolded kiwifruit were differentiated to increase sample diversity. To enhance the generalization ability of the training model results, the collected images were randomly enhanced to get 6890 kiwifruit images, and later divided into training, validation and test sets according to the ratio of 9:1:1. The final data set is shown in Table 1. The LabelImg tool, an opensource software from Github (https://github.com/heartexlabs/labelImg, accessed on 29 August 2022), was used to label the data, and the smallest outer rectangle of the fruit was used as the real frame to avoid the interference of useless pixels; the part of the kiwifruit fruit exposed in the image that was obscured or overlapped was labeled to generate the dataset file in XML format.

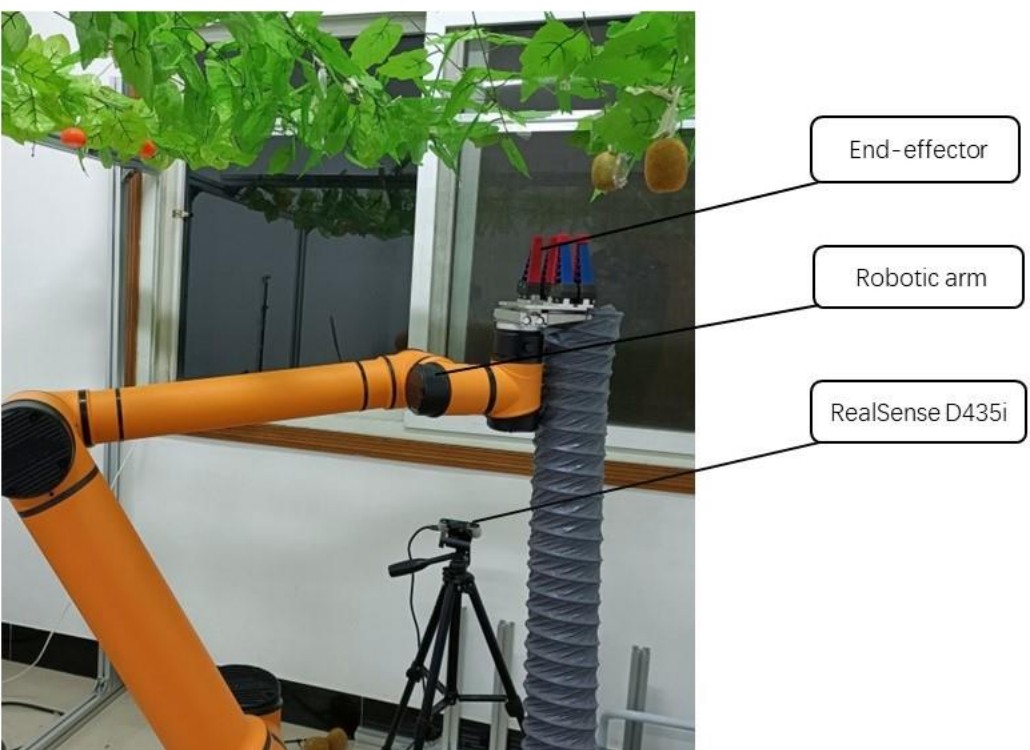

**Figure 1.** The image acquisition apparatus.

**Table 1.** The dataset of kiwifruit.

| Dataset | Occlusion | Dense | Low Light | Others |
|---|---|---|---|---|
| Training set | 1742 | 1325 | 719 | 1854 |
| Validation set | 193 | 147 | 79 | 206 |
| Test set | 193 | 147 | 79 | 206 |
| Total | 2128 | 1619 | 877 | 2266 |

### 2.2. Image Preprocessing

When a robotic vision system performs real-time monitoring of kiwifruit picking, the recognition effect is mainly affected by factors such as light intensity, robot arm vibration, branch, leaf shading situation, and fruits overlapping. In order to make the training model have better generalization ability (Figure 2), the original image is first Hue, Saturation, Value (HSV) transformed to simulate the lighting condition of scaffolded kiwifruit; the data features are enhanced using linear enhancement techniques to reduce the probability of sample inhomogeneity; to enhance the recognition of small target fruits, the image scaling is controlled and gray bars are added to the edges; padding is used to enrich the data set while avoiding learning unnecessary features; introducing Gaussian noise and pretzel noise to simulate the disturbance of the actual picking to the robot vision system and enhance the network model's ability to capture the target fruits.

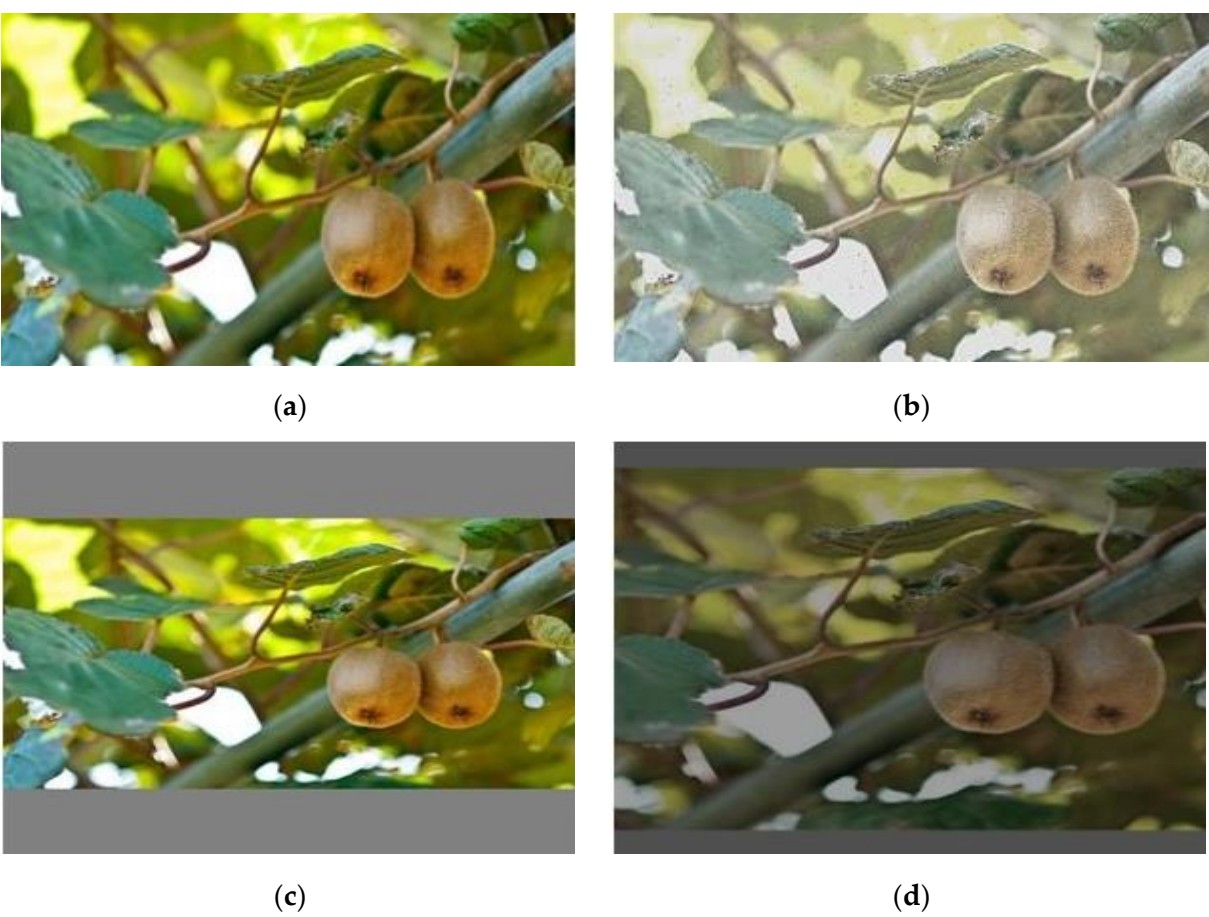

(**a**) (**b**)

(**c**) (**d**)

**Figure 2.** Manipulations of the kiwi image dataset. (**a**) Original image; (**b**) Applying salt and pepper noise; (**c**) Padding; (**d**) HSV (Hue, Saturation, Value) transforming.

The network uses the Mosaic data enhancement method to traverse four images at a time (Figure 3): firstly, the fixed area of the image is intercepted by using the matrix, and the images are inverted, scaled, and transformed by HSV color gamut; secondly, the four images are stitched into one image, and the combination of images and frames is performed, the frames beyond the image are removed, and the stitched images are edge processed; finally, the enhanced images are passed into the neural network for normalization calculation, and four images are calculated at a time, which enriches the detection background of the target fruit and speeds up the model learning efficiency.

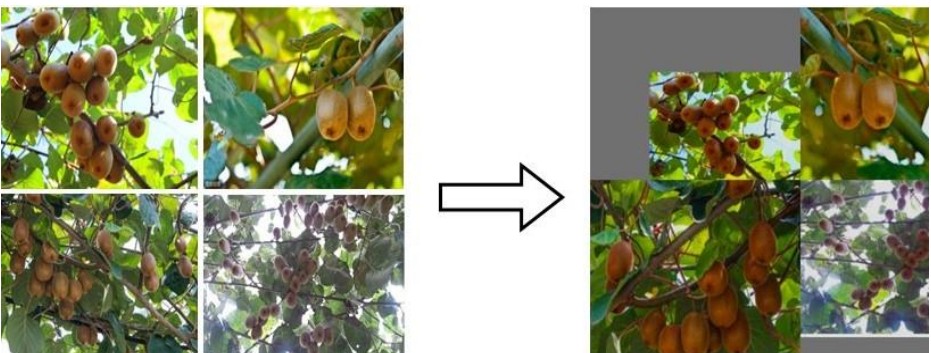

**Figure 3.** Mosaic data enhancement.

## 3. Kiwifruit Recognition Network

### 3.1. YOLOv4 Network Model

A sketch of the YOLOv4 network structure is shown in Figure 4. The network model is mainly divided into three parts: the benchmark network Backbone, Neck network, and Head output [25], which are used for classification regression and prediction through three effective feature layers. Yolo Head output: this part contains a $3 \times 3$ convolution and a $1 \times 1$ convolution, which are used for the feature set and channel number adjustment, respectively, to complete the output of target prediction results. Neck network: The SPP [26] module and PANet [27] module are used to fuse the feature information of different size feature maps to further improve the diversity and robustness of the features. SPP pools the last feature layer of the backbone network, which can greatly increase the perceptual field and separate the most significant contextual features. PANet (Path Aggregation Network) is a bottom-up feature pyramid added to the FPN (traditional feature pyramid), which achieves iterative feature extraction with strong semantic features and strong localization features. Backbone network CSPDraknet53: The CSPnet structure connects a small amount of processing directly to the end with a large residual edge, which enhances the learning ability of CNN. The Mish activation function is also introduced in the CSPnet structure, and Mish has the properties of no upper bound, fast convergence, and smooth nonmonotonicity, which helps to stabilize the network gradient flow, avoid gradient saturation, and improve the generalization ability of the model. The Mish function [28] is as Equation:

$$\text{Mish} = x \times \tan h(\ln(1 + e^x)) \tag{1}$$

where, x is the input value and $\tan h()$ is the hyperbolic tangent function.

### 3.2. YOLOv4 Object Detection Model Improvement

#### 3.2.1. Construction of the YOLOv4 Network Using the GhostNet Network

The GhostNet model is a lightweight deep network proposed by Huawei for embedded devices, whose core idea is to use less computationally intensive operations to generate redundant features, with lighter and faster features. GhostNet consists of multiple Ghost Bottleneck, and the structure of Ghost Bottleneck is shown in Figure 5. It consists of two Ghost modules and one depth-separable convolution stacked alternately with each other, and a large residual edge is formed on the other side of the stack by a $2 \times 2$ depth-separable convolution and a $1 \times 1$ normal convolution processing, which enhances the learning ability of CNN [29,30].

**Figure 4.** The structure diagram of YoloV4 network. Note: SPP stands for spatial pyramid pooling, Concat stands for tensor stitching, Conv stands for convolution, Yolo Head stands for YOLOv4 network header function.

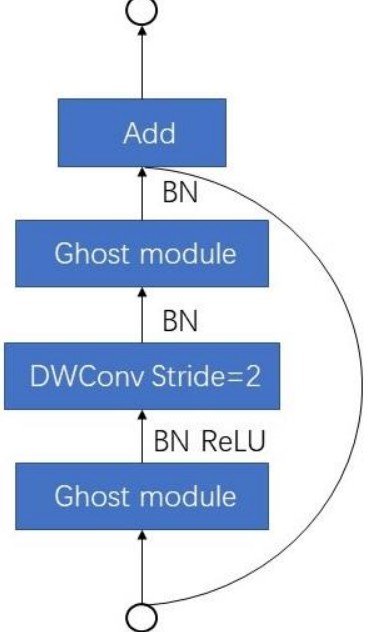

**Figure 5.** Ghost Bottlenecks.

Ghost Bottlenecks is a bottleneck structure composed of Ghost Module, the essence of which is to use Ghost Module to replace the normal convolution inside the bottleneck structure. Ghost module achieves the function of normal convolution through the combination of $1 \times 1$ convolution and depth separable convolution, which can greatly reduce the number of network parameters. The structure of Ghost Module is shown in Figure 6. The $1 \times 1$ convolution and $3 \times 3$ depth-separable convolution are used to obtain similar feature maps with dense features, which increases the perceptual field of the network and can effectively solve the problem of shallow network depth and insufficient perceptual field caused by the extensive use of $1 \times 1$ convolution in YOLOV4-tiny network. Therefore, this paper borrows this structure in the Neck network.

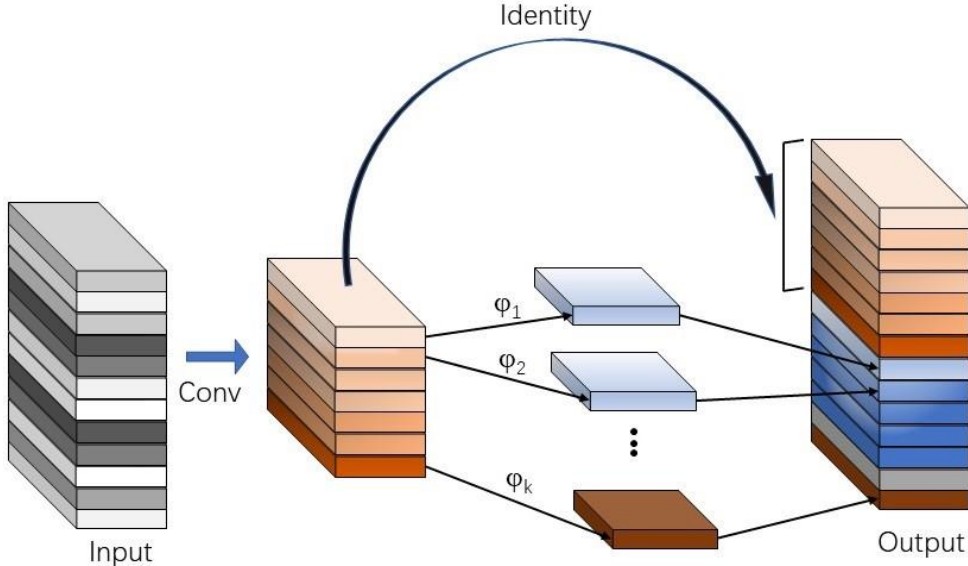

**Figure 6.** The model structure of the improved YOLOv4 network.

### 3.2.2. Improved YOLOv4 Network Model

The structure of the improved YoloV4 network model is shown in Figure 7. The feature layers with the same width and height as in CSP-Darknert53 are found from GhostNet, and these feature layers are passed into the enhanced feature network to realize the application of GhostNet in YOLOv4 network. Kiwifruit as recognition targets vary in scale, and most of them are small target fruits. The small target feature information is rough in location information and feature information is easily lost when processed by feature fusion, which causes false detection and missed detection in the network model. To improve the detection accuracy of the original model for small kiwifruit targets, $104 \times 104$ feature layers are added to aggregate the shallow feature information. Four scale feature layers of $13 \times 13$, $26 \times 26$, $52 \times 52$, and $104 \times 104$ (q4) are output from the backbone network. q4 sensory field is suitable for small object detection, and the q4 feature layer is fused with the previous feature layer by downsampling to enhance the extraction of small target feature information. Drawing on the Ghost Module network, this study proposes to use a combination of $1 \times 1$ convolution and depth-separable convolution to replace part of the normal convolution in the YOLOv4 feature extraction network, and use it as the main module to adjust the number of channels and perform inter-channel feature fusion, which can reduce the computational pressure brought by the fused feature layer q4. The Alpha parameter is introduced into the network model to replace the number of channels inside PAnet by using the parameter to adjust the parameter to improve part of the channel number adjustment in order to reduce the parameter redundancy.

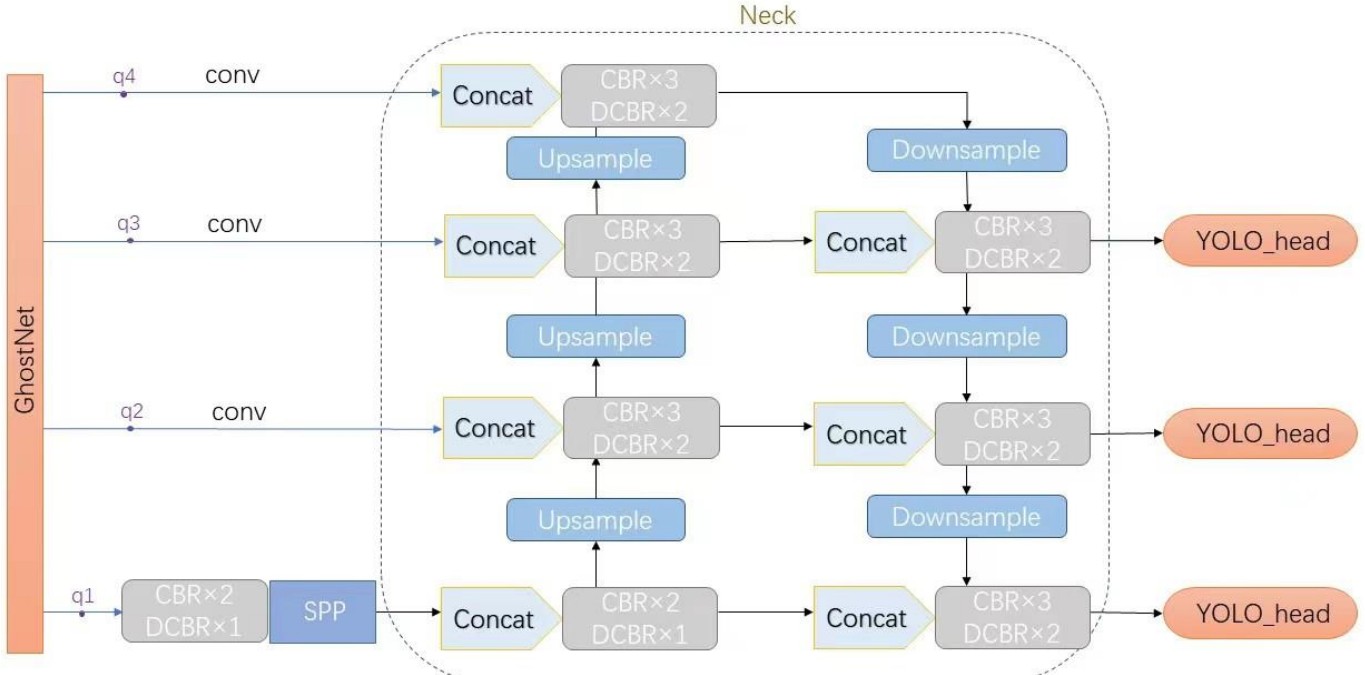

**Figure 7.** The framework of the improved YoloV4 network model. Note: Conv is convolutional, BN stands for Batch Norm, and CBR represents Conv + Batch Norm(BN) + Leaky relu activation function synthesis module, CBR means Conv + Batch Norm (BN) + Leaky relu activation function synthesis module, Depthwise Convolution plus Batch normalization plus Leaky relu activation function synthesis module, SPP stands for spatial pyramid pooling.

The loss function for network training includes regression loss function Loss(cord), confidence loss function Loss(conf) and classification loss function Loss(cls). The loss function formula is as Equations:

$$Loss = Loss(coord) + Loss(conf) + Loss(cls) \tag{2}$$

$$Loss(coord) = \lambda_{coord} \sum_{i=0}^{K \times K} \sum_{j=0}^{M} I_{ij}^{obj}(2 - I \times h_i)[L_{CIOU}] \tag{3}$$

$$Loss(conf) = -\sum_{i=0}^{K \times K} \sum_{j=0}^{M} I_{ij}^{obj}[\hat{C}_i lgC_i + (1 - \hat{C}_i)lg(1 - C_i)] - \lambda_{noobj} \sum_{i=0}^{K \times K} \times \sum_{j=0}^{M} I_{ij}^{noobj}[\hat{C}_i lgC_i + (1 - \hat{C}_i)lg(1 - C_i)] \tag{4}$$

$$Loss(cls) = -\sum_{i=0}^{K \times K} I_{ij}^{obj} \sum_{c \epsilon classes} [\hat{p}_i(c)lgp_i(c) + (1 - \hat{p}_i(c))lgI(c))] \tag{5}$$

where, K represents the grid size, I denotes the i-th square of the feature map, j denotes the j-th predicted frame of the square, w and h represent the width and height of the ground truth, respectively, obj and noobj denote the presence and absence of objects in the i-th square, respectively, $C_i$ and $\hat{C}_i$ denote the categories of predicted and true frames, respectively $p_i(c)$ is the confidence level of the predicted target, $\hat{p}_i(c)$ is the confidence level of the actual target, $\lambda_{coord}$ and $\lambda_{noobj}$ are the penalty coefficients, and $L_{CIOU}$ is the regression loss function of the bounding box.

## 4. Results and Analysis

### 4.1. Test Platform

The test platform of this paper: Windows 10, 64-bit operating system, Pytorch deep learning framework, and Python programming language. The test environment is shown in Table 2.

**Table 2.** Information about the test platform.

| Configuration | Parameter |
|---|---|
| Graphics Processing Unit (GPU) | Ge Force GTX1050Ti |
| Operating System | Windows10 |
| Accelerated Environment | Pytorch1.8.1 CUDA11.1 |
| Development Platform | Visual Studio Code |

### 4.2. Performance Metrics

In order to select a suitable model, accuracy (Precision), recall (Recall), mean average precision (mAP), average frame rate (fps), weight size (weights), and $F_1$-Score ($F_1$), are used as model performance evaluation metrics, while performance evaluation is performed using Precision-Recall curves. The calculation formula is as follows:

$$\text{Precision} = \frac{\text{TP}}{\text{TP} + \text{FP}} \tag{6}$$

$$\text{Recall} = \frac{\text{TP}}{\text{TP} + \text{FN}} \tag{7}$$

$$F_1 = 2 \times \frac{\text{Precision} \times \text{Recall}}{\text{Precision} + \text{Recall}} \tag{8}$$

$$\text{mAP} = \frac{1}{\text{C}} \sum_{\text{K}=\text{i}}^{\text{N}} \text{P(k)} \Delta \text{R(k)} \tag{9}$$

where, TP is the number of positive samples judged to be true, FP is the number of positive samples judged to be false, FN is the number of negative samples judged to be false, P(k) represents the accuracy, and R(k) represents the recall rate.

### 4.3. The Training of the Kiwifruit Recognition Network

Pre-training weights and freezing part of the training layers are used to load GhostNet weights, initial training freezes part of the network layers, and after 50 generations of training starts to unfreeze the frozen part for full network training. Putting more resources on the parameter training of the later network and unfreezing this part of the network parameters afterwards can effectively guarantee the weights. In the training process, the input image size is set to 416 × 416, the model freeze layer is set to 50 training generations, the batch sample is set to 16, the momentum factor is set to 0.9, the decay coefficient is 0.0005, and the initial learning rate is 0.001. After unfreezing, the total training generations are set to 500, the number of batch samples is eight, the momentum factor is set to 0.9, the decay coefficient is 0.0005, and the initial learning rate is 0.001. The loss value is one of the measures of the model effectiveness, and the lower the loss value is, the better the model training is theoretically. The weight file is saved once after each generation of training (epoch), and the visualization plot is recorded according to the background log information after the training is completed, and the trend of model loss value change is shown in Figure 8.

The improved YOLOv4 network model decreases the training loss value as the number of iterations increases. As can be seen from the figure, the loss function value decreases rapidly in the first 100 iteration cycles (fast model fitting); in 100–350 iterations, the loss value slowly decreases close to the optimal solution; after 70 iterations the loss value gradually stabilizes to a small fluctuation at 0.3, then the model is considered to converge.

The loss value of Ghostnet-YOLOv4 after stabilization is lower than that of MobilentV3-YoloV4, and the training results of Ghostnet-YOLOv4 network model are more satisfactory in terms of parameter convergence.

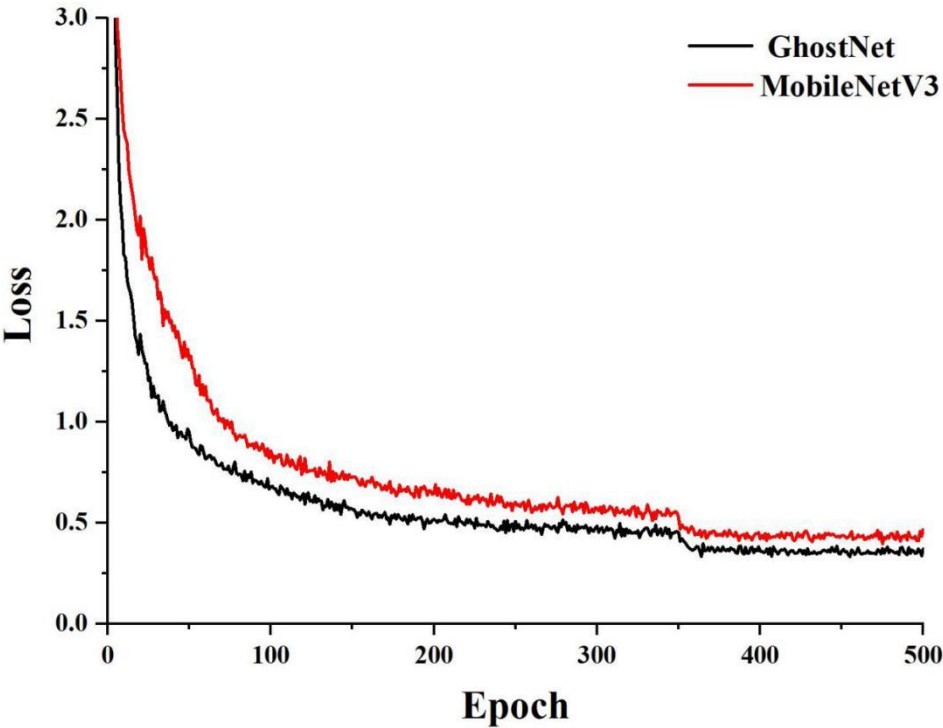

**Figure 8.** The training losses using different backbone networks.

From the loss function and Table 3, it can be seen that the loss function of the improved YOLOv4 model decreases rapidly. The corresponding average accuracy is stable at about 93.07%, which is because the improved YOLOv4 introduces a feature layer with a smaller perceptual field and improves the detection of model fine-grained. This indicates that the improved model is more capable of recognizing kiwifruit fruits in natural environments.

**Table 3.** The performances of the models using different backbone networks.

| Network Model | mAP | Recall | $F_1$ | FPS | Weights | Precision |
| --- | --- | --- | --- | --- | --- | --- |
| YOLOv4 | 91.79 | 87.94 | 87.0 | 36 | 244 | 85.26 |
| MobileNetV3-YOLOv4 | 91.44 | 86.45 | 88.0 | 47 | 53.7 | 90.04 |
| GhostNet-YOLOv4 | 93.07 | 92.43 | 92.0 | 53 | 42.5 | 90.62 |

*4.4. Improvement Results of Different Backbone Feature Extraction Networks*

The original YOLOv4 network structure is prone to target fruit under-recognition during target fruit detection due to backlighting, high fruit density, fruit overlapping, and small target fruits. The improved MobileNetV3-YOLOv4 network improves the Precision by 4.78%, F1 value by 1.0%, and detection speed by 47 frames/s compared with the original YOLOv4. The improved GhostNet-YOLOv4 improves the mAP by 5.36%, Precision by 5.36%, F1 by 5%, Recall by 4.49%, and detection speed by 53 frames/s over the original YOLOv4, with a weight reduction of 201.5 MB (Table 3). By adding feature layers favorable to small target fruit detection in the path aggregation network and introducing depth-separable convolution and Ghost Module modules, the model size can be effectively compressed, the model recognition speed accelerated, and the model recognition accuracy improved. Compared with MobileNetV3 series as the backbone network, YOLOv4 with GhostNet as the backbone network has different degrees of improvement in Precision, Recall, detection speed, and F1 value.

The two detection models were used to detect the dataset separately, and Figure 9. Shows the curves of the P and R relationships of the two networks. From the PR curves in Figure 9, the area enclosed under the PR curve of kiwifruit in the images of GhostNet-YOLOv4 model is significantly higher than that of MobileNetV3-YOLOv4. This indicated that GhostNet-YOLOv4 has higher detection accuracy and better performance.

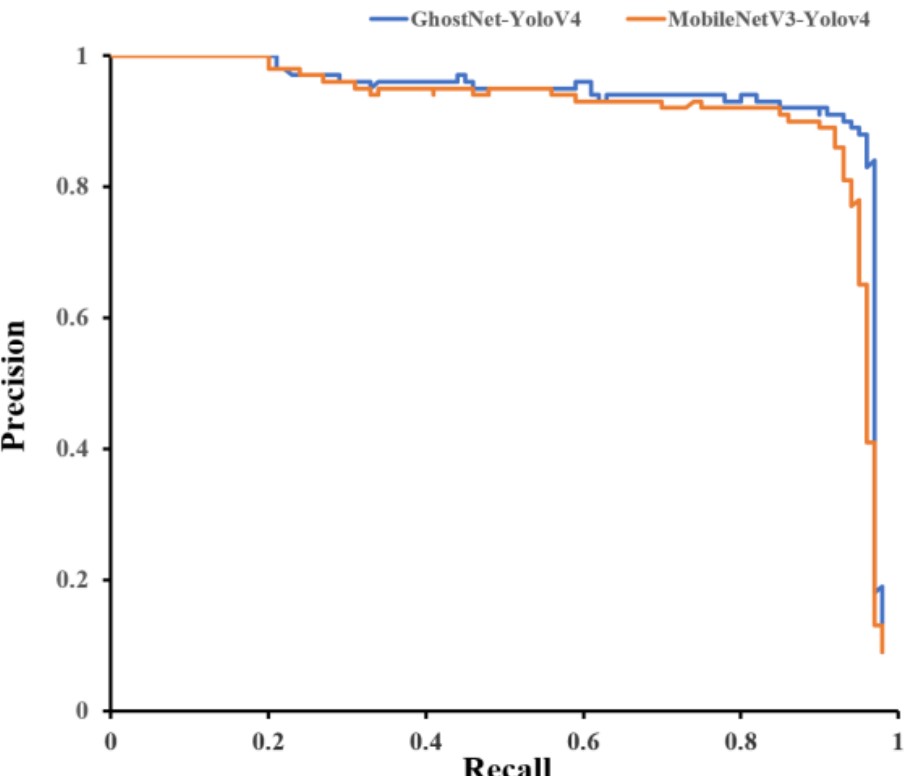

**Figure 9.** The Precision-Recall curves using different backbone networks.

### 4.5. Different Scenarios Comparison

This paper performs detection and recognition of kiwifruit for different scenes to verify the effectiveness of the improved network model. In order to visually compare the detection effect, we used the original image annotation for comparison (Figure 10). In the close-up scenes, the kiwifruit surface is evenly illuminated and detection is less difficult; in the distant scenes, the kiwifruit targets are small and dense, and the recognition accuracy of the images is lower; in the cloudy days, the light is weaker, the kiwifruit has dark shadow areas, and the features of the target fruit become blurred. The YOLOv4 model algorithm had missed detection in all cases of occlusion, dense, and insufficient illumination, and some small target fruits were not recognized, while the GhostNet-YOLOv4 detection effect was closer to the original image annotation. Compared with the YOLOv4 detection algorithm, the GhostNet-YOLOv4 model had better detection results and can effectively identify both large and small target fruits under the conditions of overcast, fruit overlap, shading and density. The improved algorithm in this study is not only applicable to images with uniform light on sunny days, but also gets better recognition results for images under low light conditions on cloudy days. The improved GhostNet-YOLOv4 network has a larger perceptual field and is more capable of recognizing kiwifruit in natural environments.

### 4.6. Comparison Experiments of Different Models

To verify the performance of the improved kiwifruit target recognition algorithm, the four object detection algorithms were evaluated by training different object detection models YOLOv3, SSD, and YOLOv4 on a desktop computer using the same kiwifruit dataset, using the best weights and the same test set for comparison tests (Figure 11). It can

be seen from Figure 11 that the detection of both YOLOv3 and SSD algorithms showed a missed recognition of the distant fruit, the main reason for this result is the small size of the distant fruit. In addition, some kiwi fruits were not recognized due to severe covered by the branches and leaves, as only a limited number of kiwifruits' features could be detected by the feature extraction network. MobileNetV3-YOLOv4 had good detection results with only a few fruit misidentifications, which were due to branch shading and insufficient light on cloudy days. As can be seen from Table 4, the mAP, Recall, and F1 value of the improved GhostNet-YOLOv4 were higher than the other object detection algorithms at an overlap threshold of 50%. The improved GhostNet-YOLOv4 object detection algorithm occupies 42.5 Mb of memory in space, while the other unimproved object detection algorithms in the comparison test had a minimum of 101 Mb, and the YOLOv4 algorithm was six times the volume of the improved algorithm. In terms of detection time, the detection speed of the YOLOv4 model was 36 frames/s, the YOLOv3 model was 41 frames/s, the SSD model was 58 frames/s, and the improved model was 53 frames/s. The mAP value of the YOLOv4 model is higher than the other three object detection models (SSD, YOLOv3, MobileNetV3-YOLOv4), but the YOLOv4 model occupies more memory, the average detection speed was slower, and the relatively large number of parameters was not recommended for porting to embedded devices. In summary, GhostNet-YOLO4 model occupied less memory and has obvious advantages in terms of detection speed and recognition accuracy.

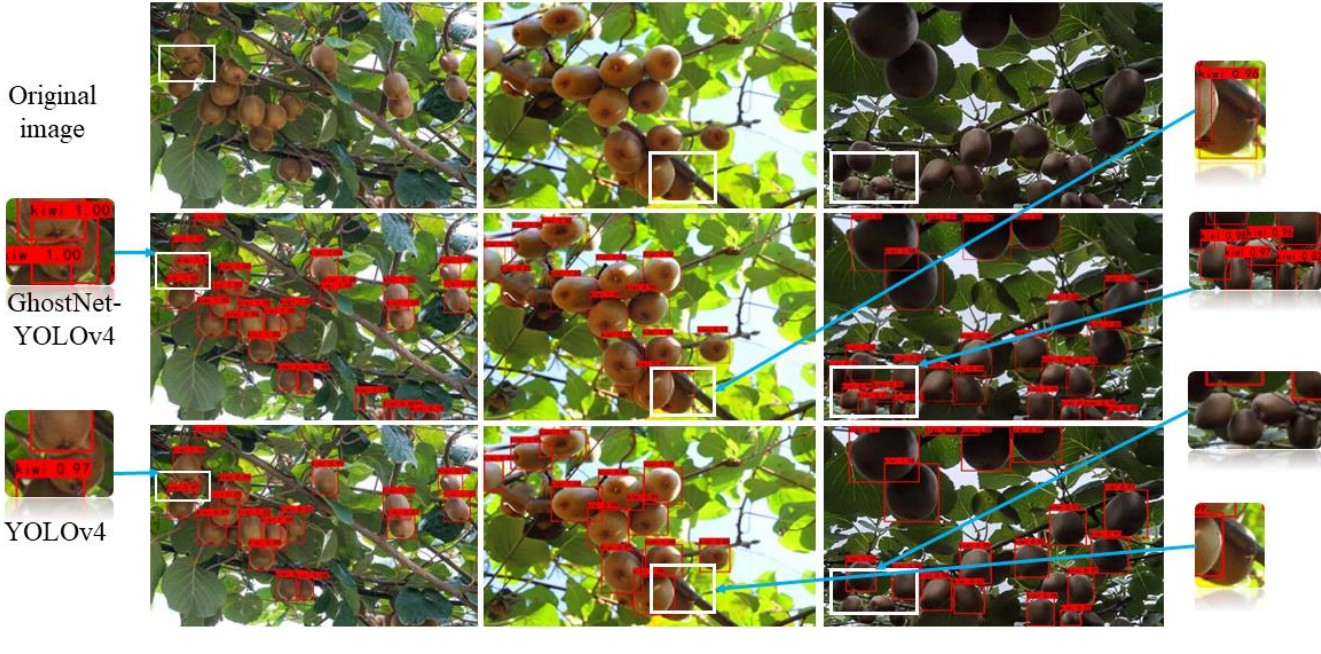

**Figure 10.** The recognition results before and after applying improvement on YOLOv4.

**Table 4.** Performances of different algorithms.

| Network Model | mAP | Recall | $F_1$ | FPS | Weights | Precision |
|---|---|---|---|---|---|---|
| MobileNetV3-YOLOv4 | 91.44 | 86.45 | 88 | 47 | 53.7 | 90.04 |
| GhostNet-YOLOv4 | 93.07 | 92.43 | 92.0 | 53 | 42.5 | 90.62 |
| YOLOv4 | 91.79 | 87.94 | 87.0 | 36 | 244 | 85.26 |
| SSD | 85.10 | 82.88 | 82.0 | 58 | 101 | 80.69 |
| YOLOv3 | 90.95 | 86.49 | 85.0 | 41 | 235 | 84.5 |

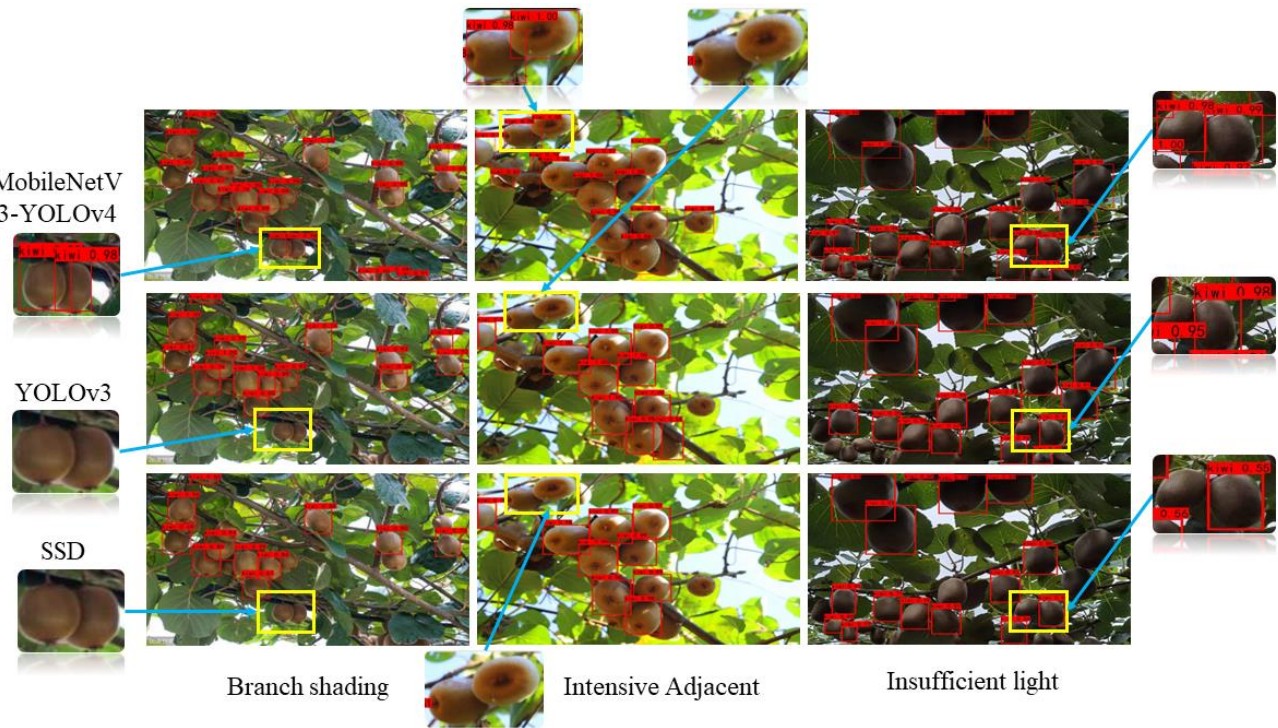

**Figure 11.** Comparison of recognition results using different models.

## 5. Conclusions

(1) In this study, we proposed an object detection model based on GhostNet to improve YoloV4 by replacing the backbone network of YoloV4 with GhostNet network, introducing feature layers adapting to small object detection in the feature fusion layer of the convolutional neural network, and using a combination of $1 \times 1$ convolution and depth-separable convolution to replace part of the normal convolution in the neck network. The improved GhostNet-YoloV4 network has better robustness with fewer weight parameters and improved the detection speed while ensuring the accuracy of kiwi recognition.

(2) The performance of the GhostNet-YoloV4 network model was evaluated, and the object detection algorithm was able to complete the recognition of kiwifruit under complex situations such as cloudy sky, shading from branches, and fruits' dense adjacency. The improved network model has a volume of 42.5 Mb, a detection speed of 42 frames/s, and an average accuracy of 93.07%, which meets the operational requirements and facilitates the application on embedded devices.

(3) Based on the actual picking environment of the orchard, an image dataset of kiwifruit was produced and the superiority of the model was verified through a pairwise comparison test. Compared with YOLOv4, GhostNet-YoloV4 compresses the network model size and improves the detection of model fine-grained by replacing the backbone network and improving the part-neck network. Using MobileNetV3_YoloV4, SSD, and YoloV4 models for testing respectively, the network model detection speed and model compression volume are better than other models with guaranteed detection accuracy.

**Author Contributions:** Data curation, J.G.; writing—original draft preparation, J.G.; visualization, X.X., L.W., Y.G. and X.S.; supervision, L.L., J.H. and S.D.; funding acquisition, M.L. All authors have read and agreed to the published version of the manuscript.

**Funding:** Key Research and Development Program of Hunan Province, Grant/Award Number: 2021SK2046; Hunan Agricultural Science and Technology Innovation Program, Grant/Award Number: 2021CX43; Hunan Agricultural Science and Technology Innovation Program, Grant/Award

**Institutional Review Board Statement:** Not applicable.

**Informed Consent Statement:** Not applicable.

**Data Availability Statement:** Not applicable.

**Conflicts of Interest:** The authors declare no conflict of interest.

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
