# Peer review of "Kiwifruit Detection Method in Orchard via an Improved Light-Weight YOLOv4"

_agronomy, doi:10.3390/agronomy12092081_

Round 1

Reviewer 1 Report

Please check the attached comments.

Author Response

Dear editor,

Thank you for editor’ and reviewers’ opinions, these comments are very helpful to improve the quality of the manuscript. We have carefully revised our manuscript, further clarify the logic of writing for improving the quality of the manuscript. Now we response the reviewers’ comments with a point by point and highlight the changes in revised manuscript. I have provide a cover letter to explain in the attachment.

  Thank you and best regards.

  Yours sincerely.

Reviewer 2 Report

This manuscript proposed Kiwifruit detection method in orchard based on a lightweight YOLOv4-GhostNet network. It is well written and covers all the required technical aspects. The result show good improvement in mAP, FPS and other performance parameters.

The authors should share the dataset and their model for the benefit of other researchers. For this GitHub or a similar platform can be used.

Author Response

(The authors gave the same response as above.)
